# Early-Life Dietary Cadmium Exposure and Kidney Function in 9-Year-Old Children from the PROGRESS Cohort

**DOI:** 10.3390/toxics8040083

**Published:** 2020-10-07

**Authors:** Edna Rodríguez-López, Marcela Tamayo-Ortiz, Ana Carolina Ariza, Eduardo Ortiz-Panozo, Andrea L. Deierlein, Ivan Pantic, Mari Cruz Tolentino, Guadalupe Estrada-Gutiérrez, Sandra Parra-Hernández, Aurora Espejel-Núñez, Martha María Téllez-Rojo, Robert O. Wright, Alison P. Sanders

**Affiliations:** 1Center for Nutrition and Health Research, National Institute of Public Health (INSP), Cuernavaca 62100, Morelos, Mexico; edds.rdz@gmail.com (E.R.-L.); carolina.ariza@insp.mx (A.C.A.); mmtellez@insp.mx (M.M.T.-R.); 2National Council of Science and Technology (CONACyT), Mexico City 03940, Mexico; 3Center for Population Health Research, National Institute of Public Health (INSP), Cuernavaca 62100, Morelos, Mexico; eduardo.ortiz@insp.mx; 4Public Health Nutrition, School of Global Public Health, New York University, New York, NY 10012, USA; ald8@nyu.edu; 5Department of Developmental Neurobiology, National Institute of Perinatology, Mexico City 11000, Mexico; ivandpantic@gmail.com; 6Department of Nutrition and Bio programming, National Institute of Perinatology, Mexico City 11000, Mexico; cruz_tolentino@yahoo.com.mx; 7Department of Immunobiochemistry, National Institute of Perinatology, Mexico City 11000, Mexico; gpestrad@gmail.com (G.E.-G.); rebe1602@hotmail.com (S.P.-H.); aurora_espnu@yahoo.com.mx (A.E.-N.); 8Department of Environmental Medicine and Public Health, Icahn School of Medicine at Mount Sinai, New York, NY 10029, USA; robert.wright@mssm.edu; 9Department of Pediatrics, Icahn School of Medicine at Mount Sinai, New York, NY 10029, USA

**Keywords:** cadmium, children, diet, kidney function

## Abstract

Cadmium (Cd) is a toxic metal associated with adverse health effects, including kidney injury or disease. The aims of this study were to estimate dietary Cd exposure during childhood, and to evaluate the association of early-life dietary Cd with biomarkers of glomerular kidney function in 9-year-old Mexican children. Our study included 601 children from the Programming Research in Obesity, Growth, Environment and Social Stressors (PROGRESS) cohort with up to five follow-up food frequency questionnaires from 1 to 9 years of age; and 480 children with measures of serum creatinine, cystatin C, and blood nitrogen urea (BUN), as well as 9-year-old estimated glomerular filtration rate. Dietary Cd was estimated through food composition tables. Multiple linear regression models were used to analyze the association between 1 and 9 years, cumulative dietary Cd, and each kidney parameter. Dietary Cd exposure increased with age and exceeded the tolerable weekly intake (TWI = 2.5 µg/kg body weight) by 16–64% at all ages. Early-life dietary Cd exposure was above the TWI and we observed inverse associations between dietary Cd exposure and kidney function parameters. Additional studies are needed to assess kidney function trajectories through adolescence. Identifying preventable risk factors including environmental exposures in early life can contribute to decreasing the incidence of adult kidney disease.

## 1. Introduction

Cadmium (Cd) is a heavy metal found naturally in the environment that has been associated with adverse health effects, such as kidney failure, bone damage, cardiovascular disease, and cancer [1]. Children’s exposure to Cd is of special concern as they may be more susceptible than adults to its toxic effects. Compared to adults, children’s food and water intake per body weight is greater, and they have increased intestinal absorption and limited renal excretion [2]. Ingestion and inhalation are the primary routes of exposure to Cd—primarily through food, tobacco smoke, and dust. Cd accumulation occurs in different tissues and organs [3]. Among the possible target organs of Cd, the kidney is one of the most sensitive organs, wherein Cd exposure is associated with tubular dysfunction, hypercalciuria, polyuria, tubulointerstitial nephritis, and low-molecular-weight proteinuria, which could lead to kidney failure in later stages [4,5,6].

There is evidence to suggest that Cd exposure plays a role in the development of chronic kidney disease (CKD) and that Cd can be nephrotoxic at environmental levels [5,7]. A recent exposure-wide association study of over 250 chemicals found that Cd was associated with CKD [8]. Therefore, identifying preventable risk factors, including environmental exposures in childhood, could contribute to our knowledgebase of early-life intervenable factors for decreasing the incidence of CKD [9]. Estimates suggest that 8–16% of the global population is affected by some form of CKD [7]. Few epidemiological studies have evaluated the association between dietary Cd and kidney function in children; however, it has been suggested that the mechanism is similar to that of adults for whom there is evidence of dietary Cd exposure and prevalence of CKD [10,11].

Regarding dietary Cd exposure, previous studies have shown elevated tolerable weekly intakes (TWI = 2.5 µg/kg body weight) in children, wherein regularly consumed food items were an important source of Cd [12,13]. A study in Uruguayan children found that Cd levels increased with age [14], and notably, a recent study in Mexican children reported an association between dietary Cd intake and urinary Cd [15]. Despite the existing evidence that children’s dietary Cd exposure is elevated, and the importance of early detection of risk factors that may affect children’s kidney health, no studies in Mexico have evaluated longitudinal dietary Cd exposure and its association with kidney function in childhood. Therefore, the aims of this study were: (1) To estimate the dietary exposure of Cd during childhood and identify the primary foods contributing to children’s Cd exposure; and (2) to evaluate the association of early-life dietary Cd exposure with biomarkers of kidney function in 9-year-old Mexican children.

## 2. Materials and Methods

### 2.1. Study Population

This analysis included children from the Programming Research in Obesity, Growth, Environment and Social Stressors (PROGRESS) cohort from Mexico City, which has been previously described in more detail [16]. Briefly, pregnant women receiving care at clinics of the Mexican Institute of Social Security were invited to participate between July 2007 and February 2011. Women were eligible if they were 18 years or older, at <20 weeks of gestation, free of heart or kidney disease, did not use steroids or anti-epilepsy drugs, did not consume alcohol on a daily basis, had access to a telephone, and planned to reside in Mexico City for the following three years.

Children were seen at our research facilities at the National Institute of Perinatology, Mexico City (INPer) at 5 follow-up visits when they were 1 year (min 0.98 years, max 1.16 years), 2 years (min 1.95 years, max 2.58 years), 4 years (min 4.00 years, max 6.75 years), 6 years (min 5.96 years, max 9.65 years), and 9 years (min 8.08 years, max 12.06 years). To estimate dietary Cd exposure, we included 601 children who had at least 1 food frequency questionnaire (FFQ) from follow-up visits. For the analysis of kidney function, 480 children with measures of creatinine, cystatin C, blood nitrogen urea (BUN), and estimated glomerular filtration rate (eGFR) at 9 years were included. We excluded children with very low birth weight (<1500 g) and/or born preterm (<37 weeks gestation) (*n* = 27). All participants provided written informed consent at the start of the study visit. The study was approved by the internal review boards of the Icahn School of Medicine at Mount Sinai (#12-00751) and the National Institute of Public Health Mexico (project #560), 31 October 2017.

### 2.2. Diet Data Collection

Children’s diet was assessed using a semiquantitative FFQ that collected food and beverage intake during the previous 7 days [17]. The questionnaire included 101 foods grouped into 14 categories (dairy products, fruits, vegetables, fast food, meat, fish, legumes, cereals, corn products, beverages, snacks, soups, miscellaneous, and tortillas). Frequency values ranged from never consumed, to consumed 7 days per week, as well as times per day that they were consumed with values ranging from consumed 1 to 6 times per day. Serving size (small, medium, large, and very large) and the number of servings consumed were reported. The questionnaire was administered by an interviewer at each of the study visits and answered by the child’s mother or caregiver.

#### Dietary Cadmium

To estimate dietary Cd, we carried out a search of composition tables reporting Cd levels in food, based on the Total Diet Study (TDS) methodology, which consists of reporting concentrations of contaminants in food [18]. As Mexico lacks such tables, we used tables from the United States, The European Food Safety Authority, Australia, Hong Kong, and Canada. We also used available Cd concentrations measured in selected meat products (ham, sausage, and chorizo) by a study in Mexico City [19]. Each food and beverage item in our FFQ was matched with each of the items found in the Cd composition tables of the different countries. We used the composition tables to obtain an average Cd concentration for each food item, i.e., when the food item was included in all five tables, the average Cd concentration from the tables was used, and so on, if the food item was included in only one table that value was used. In some cases, more than one food item was inquired as the same item on our FFQ (e.g., “broccoli and cauliflower”); for those cases, we used the average Cd for each of the food items from the composition tables. Cd from dishes and prepared foods was estimated (per 100 g of preparation) using a standardized method that accounts for grams of each ingredient in the recipe. Cd levels were estimated in µg/day and were reported for each study visit, as well as the “top ten” foods that contribute the most to the Cd intake according to their frequency of consumption.

We also created a dichotomous variable for children’s Cd exposure at each study visit according to the European Food Safety Authority TWI for Cd of 2.5 µg/kg body weight. If the child’s Cd intake was above the TWI: High = 1, or below the TWI: Low = 0 [20]. Finally, to account for approximate cumulative Cd exposure, we generated an ordinal Cd score by adding the Cd TWI high or low from all study visits (scores ranged from 0 = low exposure in all study visits, to 5 = high exposure in all study visits). This was done only for children who had dietary information for all five study visits (*n* = 182).

### 2.3. Kidney Function Parameters

Kidney function parameters were determined at 9 years by standardized and trained staff in the Nutrition and Bio-programming Research, and the Immunochemistry department’s laboratories at INPer. Fasting blood samples were collected in BD Vacutainer tubes, and serum was separated according to the standard protocol and stored at −70 °C until the analysis. The laboratory analyses were carried out using the following methods:

*Serum creatinine (SCr)* was measured through the kinetic test without deproteinization according to the Jaffé method [21]. *Cystatin C (Cys C)* was measured by Quantikine Human Cystatin C enzyme-linked immunosorbent assay. *Blood urea nitrogen* (*BUN)* was calculated with the following formula: Serum urea (mg/dL)/2.14; serum urea was determined through the Urease—GLDH test: Enzymatic UV test [22]. *Estimated glomerular filtration rate* (*eGFR)* was calculated using two formulae: (1) Schwartz formula: eGFR_Schwartz_ = (*k* × height)/SCr, where *k* is 0.55 for children under 13 years, height is measured in cm, and SCr is in mg/dL [23]; and (2) the 2012 Cystatin C-based equation: eGFR_Cystatin C_ = 70.69 × (CysC)^−0.931^, where cystatin C is in mg/L [24].

### 2.4. Covariates

Information on secondhand tobacco exposure was obtained by questionnaire; mothers reported the minutes per day that children spent with smokers at 4, 6, and 9 years. Children were weighed with the least amount of clothing possible and without footwear using an InBody230, and height was measured with a SECA stadiometer without footwear. The z-score for body mass index (BMI) was estimated according to WHO guidelines [25]: Underweight, normal weight, overweight, and obesity were defined as <−2SD, >−2SD to ≤+1SD, >+1SD to +2SD, and >+2SD, respectively. As there were few underweight children, these observations were collapsed to the normal weight category. Physical activity was measured with the International Physical Activity Questionnaire (IPAQ) that was answered by the child’s mother at 4, 6, and 9 years. We considered moderate to vigorous activities such as play in the park, run, walk, ride a bike, and dance to estimate minutes of aerobic activities per day. Maternal socioeconomic status was collected at the time of enrollment using a questionnaire according to the Mexican Association of Market Intelligence and Public Opinion Agencies (AMAI, 2007 version). The AMAI classifies Mexican households into seven levels (very low, low, middle-low, middle, middle-high, and high) according to their ability to satisfy the needs of their members. In this study, we collapsed the AMAI levels into three categories: lower, middle, and higher.

### 2.5. Statistical Analysis

We performed descriptive analyses to identify the food items at each study visit that most contributed to the estimated total dietary Cd intake and reported them as percentages.

We analyzed the distributions and descriptive statistics for each kidney parameter and log-transformed variables for cystatin C, BUN, and eGFR. We used multivariable linear regression models to analyze associations between dietary Cd at 1 and 9 years with kidney parameters at 9 years. In a subset of children with dietary information across all five study visits, we derived an ordinal cumulative Cd score. The ordinal score was derived as follows: A score of 0 indicated no exposure throughout the study visits, a score of 1 indicated a high dietary Cd in one study visit and so forth, and 5 indicated high dietary Cd in all ages (i.e., 1, 2, 4, 6, and 9 years). We incorporated the ordinal score as a discrete and categorical variable in models and examined associations with each kidney function parameter. Final models were adjusted for sex, age in months (to account for the wider age range at the study visit), BMI z-score, physical activity, secondhand tobacco exposure, and socioeconomic status. All statistical analyses were performed in Stata Statistical Software: Release 14. College Station, TX: StataCorp LP.

## 3. Results

Table 1 shows the participant characteristics at each study visit. Just over half of the children were male, and the age ranged from 12.2 ± 0.28 months at the 1-year visit to 116 ± 8.2 months at the 9 year visit. Most of the children were of normal weight between 1 and 6 years; however, at 9 years, almost half of the children were overweight (24.8%) or obese (22.0%). Physical activity at 4 and 6 years was 65.8 ± 30.0 and 69.73 ± 29.3 min/day on average, respectively, and decreased to 21.0 ± 7.10 min/day at 9 years. The majority of children (more than 80% at 4, 6, and 9 years) were not exposed to secondhand tobacco smoke and were of middle or lower socioeconomic status.

Table 2 shows kidney function parameters measured at 9 years; the mean cystatin C was 0.730 ± 0.17 mg/dL, while the mean BUN was 12.20 ± 3.09 mg/dL. Using the eGFR_CystatinC_ formula, there were two children below 60 mL/min/1.73 m^2^ and 75 children in the 60–90 mL/min/1.73 m^2^ range, while for the eGFR_Schwartz_, a single child was between 60 and 90 and the rest had and eGFR ≥ 90 mL/min/1.73 m^2^. We note that while fewer children had SCr measures, this observation is in line with evidence that SCr-based equations may overestimate eGFR [26].

Figure 1 shows the primary foods contributing to the estimated dietary intake of Cd from 1 to 9 years of age. At 1 and 2 years, the top contributors were leafy greens (1 year: 16.0%, 2 years: 9.0%), milk (1 year: 10.1%, 2 years: 9.8%), and carrots (1 year: 8.8%, 2 years: 7.9%); at 4 years, the top contributors were sweets (6.8%), milk (6.1%), and carrots (4.8%); at 6 years, the top contributors were lettuce (6.8%), sandwich (6.6%), and sweets (6.4%); and at 9 years, the top contributors were lettuce (6.0%), pasta soup (5.7%), and sweets (5.5%). Total dietary Cd intake was 4.43 ± 2.53 µg/d at 1 year; 4.65 ± 2.45 µg/d at 2 years; 6.00 ± 3.45 µg/d at 4 years; 6.83 ± 3.15 µg/d at 6 years; and 8.09 ± 4.33 µg/d at 9 years. According to the TWI, Cd intakes were exceeded by children at all study visits (study visit, % children): 1 year, 64%; 2 years, 49%; 4 years, 35%; 6 years, 28%; and 9 years, 16%. For the cumulative Cd score (*n* = 175 children with information on dietary Cd in all study visits), we saw that 23 children had low Cd across all study visits; 38 children had high Cd in one study visit; 57 children had high Cd in two study visits; 31 children had high Cd in three study visits; 16 children had high Cd in four study visits; and 10 children had high Cd intake in all five study visits.

For the cross-sectional associations between higher 9-year dietary Cd and kidney parameters (Table 3), we observed an inverse association between dietary Cd and BUN (β = −0.077 (95% CI: [−0.151, −0.003])) and, marginally, with eGFR_CystatinC_ (β = −0.046 (95% CI: [−0.107, 0.014])).

We observed no significant associations between 1-year dietary Cd and subsequent children’s kidney function (Table 4).

Finally, we assessed associations with cumulative Cd intake using an ordinal score, analyzed as discrete and categorical (Table 5). We again identified an inverse association with BUN when using the score as a discrete variable; however, no associations with the other parameters were observed. Using the score as categorical, we observed an inverted “U-shaped” relationship with cystatin C where estimates decreased with an increasing cumulative Cd score to 3 and then increased with the cumulative Cd score from 4 and 5; and a negative dose-response for BUN where, as the score increased, the model estimates decreased progressively from −0.006 to −0.187 (Table 5).

## 4. Discussion

In this longitudinal study of 601 Mexican children, we found that the estimated dietary Cd exposure increased with age, from ~4.4 µg/d at 1 year to 8.1 µg/d at 9 years. The main food sources of Cd changed across all study visits; at 1 and 2 years, leafy greens, milk, and carrots were the primary dietary contributors, and beginning at 4 years, we observed a dietary transition to sweets, lettuce, and sandwiches as the primary contributors. This transition most likely reflects maternal control of the child’s dietary intake during the first two years of life and then the child’s food preferences influencing intake at older ages. TWI was exceeded at all study visits, with 64% at 1 year decreasing to 16% in 9-year children consuming more Cd than is recommended. Finally, we found inverse associations between dietary 9-year high Cd intake, as well as the cumulative score and BUN β = −0.077 (95% CI: (−0.151, −0.003)) and β = −0.037 (−0.072, −0.003), respectively.

Our results are similar to the Cd intake observed among 4–12-year-old Australian children where the Cd intake was 4.0 ± 2.2 (0.98–9.5) µg/d [27] but higher than a study from the United States where the mean Cd intake in 2 to 10-year-old children was 2.96 (2.83, 3.10) µg/d [28]. Regarding the main food contributors, these results are similar to a previous study in the United States where the top Cd contributors to children’s diet at 10–11 years old were milk, lettuce, and cookies [28]. The TWIs are in line with a study in France that observed that diets of 5–6-year-old children exceeded the Cd TWI by 12–15% [12], while diets of 1 to 3-year-old and 6-year-old Finnish children exceeded the TWI by 88% and 64%, respectively [13].

Cd intake was comparable to other populations, suggesting that at these daily intakes, the kidney parameters we studied are not altered in 9-year-old children. There are several possible interpretations of our findings. It is possible that at this early stage of childhood (age 8–12), potential changes in kidney function are not yet apparent. This is likely given that glomerular function, as assessed via eGFR and less specifically with BUN, may not show a marked decrease until substantial dysfunction has occurred [29]. Thus, we note with interest the marginally significant inverse associations with eGFR as early as age 8–12 observed in this study. The earliest indication of kidney damage in humans is typically an increased excretion of low-molecular-weight proteins, such as β2-microglobulin, α1-microglobulin, retinol binding protein, and N-acetyl-β-glucosaminidase, among others; and increased excretion of calcium and metallothionein [30]. We also note that BUN is a nonspecific biomarker that can vary independently of the GFR. BUN can be associated with other factors such as liver dysfunction [31], and low or high BUN may be related to undesirable states like malnutrition, starvation, dehydration, high protein intake, among others [32]. Future studies should assess biomarkers of subclinical injury, including urinary proteins like β2-microglobulin and additional assessment of liver function.

Another interpretation is that there is no kidney damage at these Cd intake concentrations. We would not anticipate ‘normal’ dietary levels of Cd to cause clinically apparent kidney dysfunction. However, there is evidence that when Cd reaches between 50 and 300 μg/g wet weight in the kidney cortex, the amount of Cd not bound to metallothionein becomes sufficiently high to cause tubular damage [30]. Studies have estimated that to reach these Cd levels, a Cd intake of >200 µg/day or lifetime intake of 1300 mg is necessary [33]. Lastly, it is possible that a compensatory mechanism exists, where kidney function in healthy children can cope with this level of Cd insult, while high levels of Cd are nephrotoxic [34].

An important limitation of our study is the estimation of dietary Cd using FFQs, as these can introduce measurement error from maternal recall. It is unlikely though that this could have led to bias as mothers were unaware of the research question (i.e., did not answer the questionnaire in terms of Cd exposure or kidney function). Nonetheless, FFQs are also subject to measurement error as, beyond maternal recall, children spend time in school or in other contexts where they consume food items most likely not reported in the FFQ. Furthermore, we did not measure the concentrations directly in food items. Except for Cd concentrations for a few food items (processed meat) reported in a study of adults in Mexico City [19], we used data from nutrient composition tables from different countries, where the Cd concentrations reported for food items may be different from Mexico. Ideally, our study would have measured the Cd concentrations in food items reported in the FFQ and purchased samples in local markets; however, this was beyond the scope of this study. By using food content tables from the United States, The European Food Safety Authority, Australia, Hong Kong, and Canada, we aimed at having a higher diversity of food items, as well as a higher variability for the Cd concentrations, also considering the reality of globalized food supply chains and the availability of international food products in Mexican markets.

We therefore cannot rule out that we either under- or over-estimated Cd intake; however, the results are in line with levels reported in previous studies. For example, a study conducted by the United States Food and Drug Administration showed that the top contributors to dietary Cd were grains, prepared foods (e.g., hamburgers, pizza, lasagna, soups), and vegetables [2]; a study of French children showed that the main contributors to dietary Cd were bread and potatoes [35]; and a study of Chinese children found that the three greatest contributors to dietary Cd were rice, leafy vegetables, and wheat flour [36]. Further, Cd intake estimated by FFQ does not reflect its absorption, metabolism, or excretion. Cd absorption differs between individuals based on their nutritional status, particularly levels of zinc and iron. For this study, we lacked data on indicators of children’s nutritional status of these micronutrients. We also lacked urinary Cd measurement, which is an indicator of body burden [37]. PROGRESS has archived urine samples in which Cd concentrations will be analyzed in future studies.

Among the strengths of this study are the repeated measurements of diet at five stages ranging from 1 to 9 years of age. An important observation of this study was the marked increase in children with overweight or obesity; at 4 years, only 18% of the children presented with overweight or obesity, but by age 9 years, this increased to 46.8%. We also observed an important decrease in physical activity between 6 and 9 years. Both obesity and physical activity are important metabolic risk factors that could modify the association between Cd exposure and kidney parameters [33,38]. Future studies will directly assess the role of obesity and physical activity as risk factors for longitudinal childhood kidney function trajectories.

Although we did not observe significant associations between children’s Cd intake and kidney parameters, the TWI for Cd was exceeded at all study visits, from 1 to 9 years of age. The TWI for Cd is based on preventing downstream effects on kidney, bone, and cardiovascular health in adulthood because it can accumulate throughout life. Follow-ups of this study population at later life stages will elucidate possible Cd nephrotoxicity.

## Figures and Tables

**Figure 1 toxics-08-00083-f001:**
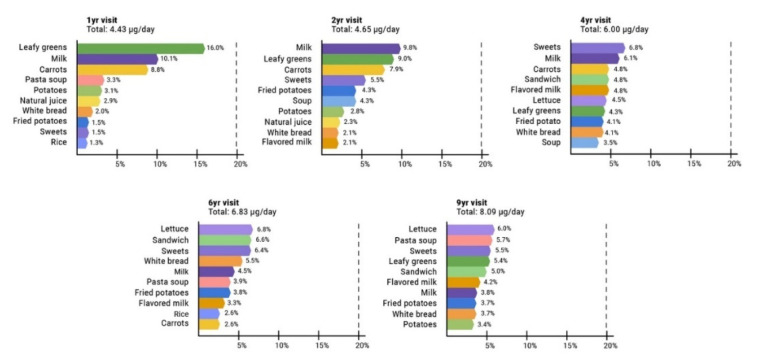
Primary foods contributing to estimated dietary Cd intake in 1–9-year-old children.

**Table 1 toxics-08-00083-t001:** Participant characteristics at follow-up study visits.

Study Visit	1-Year Visit *n* = 566	2-Year Visit *n* = 530	4-Year Visit *n* = 582	6-Year Visit *n* = 573	9-Year Visit *n* = 544
Characteristic	Mean ± SD or *n* (%)	Mean ± SD or *n* (%)	Mean ± SD or *n* (%)	Mean ± SD or *n* (%)	Mean ± SD or *n* (%)
Sex Male	285 (50.3%)	278 (52.4%)	293 (50.3%)	293 (51.2%)	280 (51.4%)
Age (months)	12.2 ± 0.28	24.4 ± 0.54	58.5 ± 6.7	82.1 ± 7.2	116 ± 8.2
BMI Z-score ^a^					
Underweight	15 (2.6%)	2 (0.38%)	2 (0.34%)	7 (1.2%)	3(0.53%)
Normal weight	455 (80.3%)	352 (66.4%)	474 (81.4%)	407 (71.0%)	286 (52.5%)
Overweight	73 (12.9%)	130 (24.5%)	68 (11.6%)	92 (16.0%)	135 (24.8%)
Obesity	23 (4.0%)	46 (8.6%)	38 (6.5%)	67 (11.6%)	120 (22.0%)
Physical activity (min/day) ^b^ Aerobic activities	N/A	N/A	65.8 ± 30.0	69.73 ± 29.3	21.02 ± 7.10
Second-hand smoking (daily) ^c^	N/A	N/A			
Yes			58 (10%)	75 (13%)	98 (18.1%)
No			517 (90%)	486 (86.6%)	442 (81.8%)
Socioeconomic Status ^d^					
Lower	292 (51.5%)				
Middle	218 (38.5%)				
Higher	56 (9.8%)				

^a^: Classification according to BMI/age z-score (Underweight <−2SD, normal >−2SD to ≤+1SD, overweight >+1SD to +2SD, obesity >+2SD); ^b^: Aerobic activities: Play in the park, run, walk, ride a bike, dance, stage 48−*n* = 7 missing observations; ^c^: Classification according to the minutes that children expend with smokers, stage 48−*n* = 7 missing observations, stage 72−*n* = 12 missing observations, stage 96−*n* = 4 missing observations; ^d^: Classification according to the AMAI index; data from the enrollment.

**Table 2 toxics-08-00083-t002:** Kidney function parameters from 9-year-old children.

Kidney Function Parameter	*n*	Mean ± SD	(5–95%)
Serum Creatinine (mg/dL)	380	0.434 ± 0.09	(0.28–0.59)
Cystatin C (mg/L)	455	0.730 ± 0.17	(0.48–1.03)
BUN (mg/dL)	379	12.20 ± 3.09	(7.71–17.9)
eGFR_Schwartz_ (ml/min/1.73 m^2^)	379	180.91 ± 41.65	(125.43–261.8)
eGFR_Cystatin C_ (ml/min/1.73 m^2^)	455	116.55 ± 28.41	(77.93–166.79)

**Table 3 toxics-08-00083-t003:** Association between tolerable weekly intake (TWI) ^a^ dietary cadmium exposure at 9 years and concurrent kidney function parameters in children.

			Unadjusted		Adjusted ^b^
		Low CdD	High CdD		High CdD
	*n*		β (95% CI)	*n*	β (95% CI)
SCr ^c^ (mg/dL)	375	Ref	−0.021 (−0.047, 0.005)	342	−0.013 (−0.041, 0.015)
Cystatin C (mg/L)	447	Ref	0.057 (0.001, 0.113)	409	0.049 (−0.010, 0.109)
BUN (mg/dL)	375	Ref	−0.038 (−0.111, 0.033)	342	−0.077 (−0.151, −0.003)
eGFR_Schwartz_ (ml/min/1.73m^2^)	376	Ref	0.020 (−0.043, 0.084)	343	0.022 (−0.046, 0.091)
eGFR_Cystatin C_ (ml/min/1.73m^2^)	449	Ref	−0.053 (−0.110, 0.004)	411	−0.046 (−0.107, 0.014)

^a^ Tolerable Weekly Intake: High ≥2.5 µg/kg body weight, low: <2.5 µg/kg body weight. ^b^ adjusted for sex, age, z-score BMI, physical activity, secondhand smoke, and socioeconomic status. ^c^ Serum creatinine.

**Table 4 toxics-08-00083-t004:** Association between TWI dietary cadmium exposure (high vs. low) ^a^ at 1-year and kidney function parameters in 9-year-old children.

			Unadjusted		Adjusted ^b^
		Low CdD	High CdD		High CdD
	*n*		β (95% CI)	*n*	β (95% CI)
SCr ^c^ (mg/dL)	229	Ref	−0.012 (−0.039, 0.013)	208	−0.003 (−0.031, 0.025)
Cystatin C (mg/L)	274	Ref	−0.051 (−0.110, 0.007)	248	−0.026 (−0.089, 0.036)
BUN (mg/dL)	229	Ref	−0.055 (−0.125, 0.014)	208	−0.011 (−0.081, 0.059)
eGFR_Schwartz_ (ml/min/1.73m^2^)	230	Ref	0.027 (−0.035, 0.090)	209	−0.003 (−0.072, 0.065)
eGFR_Cystatin C_ (ml/min/1.73m^2^)	276	Ref	0.065 (0.004, 0.125)	250	0.034 (−0.029, 0.098)

^a^ Tolerable Weekly Intake: High ≥2.5 µg/kg body weight, low: <2.5 µg/kg body weight. ^b^ adjusted for sex, age, z-score BMI, physical activity, secondhand smoke, and socioeconomic status. ^c^ Serum creatinine.

**Table 5 toxics-08-00083-t005:** Association between estimated cumulative dietary Cd score (as a discrete and categorical variable) and 9-year kidney function parameters ^a^.

Kidney Function Parameter	SCr ^b^ (*n* = 124)	Cystatin C (*n* = 140)	BUN (*n* = 124)	eGFR_Schwartz_ (*n* = 124)	eGFR_CystatinC_ (*n* = 142)
Score	β (95% CI)	β (95% CI)	β (95% CI)	β (95% CI)	β (95% CI)
Discrete	−0.011 (−0.025, 0.002)	−0.019 (−0.050, 0.010)	−0.037 (−0.072, −0.003)	0.025 (−0.007, 0.058)	0.020 (−0.011, 0.051)
0 (*n* = 23)	Ref	Ref	Ref	Ref	Ref
1 (*n* = 38)	0.017 (−0.047, 0.081)	0.063 (−0.083, 0.211)	−0.006 (−0.168, 0.154)	−0.028 (−0.182, 0.126)	−0.064 (−0.216, 0.088)
2 (*n* = 57)	−0.015 (−0.075, 0.044)	0.053 (−0.087, 0.193)	−0.041 (−0.191, 0.107)	0.048 (−0.094, 0.190)	−0.072 (−0.216, 0.071)
3 (*n* = 31)	−0.039 (−0.0105, 0.026)	−0.082 (−0.241, 0.075)	−0.092 (−0.257, 0.073)	0.104 (−0.053, 0.262)	0.081 (−0.082, 0.244)
4 (*n* = 16)	0.000 (−0.081, 0.081)	−0.028 (−0.209, 0.151)	−0.129 (−0.332, 0.073)	0.011 (−0.182, 0.204)	0.027 (−0.159, 0.213)
5 (*n* = 10)	−0.051 (−0.147, 0.044)	−0.007 (−0.202, 0.187)	−0.187 (−0.428, 0.052)	0.100 (−0.128, 0.328)	0.009 (−0.192, 0.210)

^a^ adjusted for sex, age, z-score BMI, physical activity, secondhand smoke, and socioeconomic status. ^b^ Serum creatinine.

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
