# Peer review of "Early-Life Dietary Cadmium Exposure and Kidney Function in 9-Year-Old Children from the PROGRESS Cohort"

_toxics, 2020, doi:10.3390/toxics8040083_

Round 1

Reviewer 1 Report

Minor corrections to grammar/spelling needed.

The word kidney is preferable to renal.

Urinary concentrations of  of Cd should have been determined.

Limitation of this study have been acknowledged

Author Response

Minor corrections to grammar/spelling needed. We have carefully checked the grammar/spelling for the manuscript. Dr. Sanders and I have reviewed the manuscript carefully.   

The word kidney is preferable to renal. We have changed the word “renal” to “kidney” throughout the manuscript.

Urinary concentrations of Cd should have been determined. This is an important comment, unfortunately at the time of this study the Cd concentrations in urine for 9 year old children have not been analyzed. We will certainly follow-up once urine analyses are complete (e.g., at ages 6 and 9) in a future study.  

We have added this text in the manuscript (page 8, line 303):

“PROGRESS has archived urine samples in which Cd concentrations will be analyzed and results included in future studies”.

Limitation of this study have been acknowledged. Thank you for this comment, we have added a few paragraphs to the discussion based on the comments of Reviewer 2.

Reviewer 2 Report

Data for these analyses derive from PROGRESS, a longitudinal study of children in Mexico. The goal was to determine if cadmium exposure during childhood, estimated five times between age 1 and 9 years, was associated with multiple biomarkers of kidney function at age 8-9 years. Cadmium exposure estimated using food composition derived from food frequency questionnaire between ages 1 and 9 years and outcome data of serum creatinine, cystatin C, and blood nitrogen urea (BUN), glomerular filtration rate, was available for 480 children.  

This is a well written manuscript and presents important findings showing that 16-64% of children exceeded the tolerable weekly intake of cadmium, and that dietary cadmium was associated some of the measures of kidney function in children. A strength of the manuscript is the use of multiple indicators of kidney function, as individually, these biomarkers may not be as specific in children. Identifying potential sources of cadmium in the diet is also a major strength of this manuscript.

The major weakness of the manuscript is reliance on dietary reports to estimate cadmium exposure. First, food frequency questionnaires are inaccurate as they require recall of food items consumed in the past, and this problem is exacerbated by reliance on adults, e.g., parents to providing the data for their children. Second, the use of food tables to estimate cadmium assumes a level of contamination in food items reported to have been consumed, yet in a globalized food supply chain, this may or may not be the case.

While the findings are still intriguing, the effects of these weaknesses on the results reported , warrant, at a minimum a discussion. That is, it requires a prominent place in the discussion section of the manuscript.

Author Response

Reviewer 2:

The major weakness of the manuscript is reliance on dietary reports to estimate cadmium exposure. First, food frequency questionnaires are inaccurate as they require recall of food items consumed in the past, and this problem is exacerbated by reliance on adults, e.g., parents to providing the data for their children.

We thank the Reviewer for making this point, we have added the following to the paragraph addressing this issue (page 8, line 278):

“An important limitation of our study is the estimation of dietary Cd using FFQs, as these can introduce measurement error from maternal recall. It is unlikely though that this could have led to bias since moms were unaware of the research question (i.e. did not answer the questionnaire in terms of cadmium exposure or kidney function). Nonetheless, FFQs are also subject to measurement error since, beyond maternal recall, children spend time in school or in other contexts where they consume food items most likely not reported in the FFQ”.

Second, the use of food tables to estimate cadmium assumes a level of contamination in food items reported to have been consumed, yet in a globalized food supply chain, this may or may not be the case.

We agree with the reviewer’s point and have added the following text to address this (page 8, line 283; page 3, line 112):

“Furthermore, we did not measure the concentrations directly in food items. Except for Cd concentrations for a few food items (processed meat) reported in a study of adults in Mexico City [20], we used data from nutrient composition tables from different countries, where the Cd concentrations reported for food items may be different from Mexico. Ideally our study would have measured the Cd concentrations in food items reported in the FFQ and purchased samples in local markets, however this was beyond the scope of this study. By using food content tables from the United States, The European Food Safety Authority, Australia, Hong Kong and Canada we aimed at having more diversity of food items as well as more variability for the Cd concentrations, also considering the reality of globalized food supply chains and the availability of international food products in Mexican markets”.

“We also used available Cd concentrations measured in selected meat products (ham, sausage and chorizo) by a study in Mexico City [20]”.

While the findings are still intriguing, the effects of these weaknesses on the results reported, warrant, at a minimum a discussion. That is, it requires a prominent place in the discussion section of the manuscript.

We believe we have now addressed the comments adequately.
